# INSTRUCTED DIFFUSER WITH TEMPORAL CONDITION GUIDANCE FOR OFFLINE REINFORCEMENT LEARNING

## ABSTRACT

Recent works have shown the potential of diffusion models in computer vision and natural language processing. Apart from the classical supervised learning fields, diffusion models have also shown strong competitiveness in reinforcement learning (RL) by formulating decision-making as sequential generation. However, incorporating temporal information of sequential data and utilizing it to guide diffusion models to perform better generation is still an open challenge. In this paper, we take one step forward to investigate controllable generation with temporal conditions that are refined from temporal information. We observe the importance of temporal conditions in sequential generation in sufficient explorative scenarios and provide a comprehensive discussion and comparison of different temporal conditions. Based on the observations, we propose an effective temporally-conditional diffusion model coined Temporally-Composable Diffuser (TCD), which extracts temporal information from interaction sequences and explicitly guides generation with temporal conditions. Specifically, we separate the sequences into three parts according to time expansion and identify historical, immediate, and prospective conditions accordingly. Each condition preserves non-overlapping temporal information of sequences, enabling more controllable generation when we jointly use them to guide the diffuser. Finally, we conduct extensive experiments and analysis to reveal the favorable applicability of TCD in offline RL tasks, where our method reaches or matches the best performance compared with prior SOTA baselines.

## 1 INTRODUCTION

Diffusion probabilistic models (DPMs) have shown impressive results in photo-realistic image synthesization (Ho et al., 2020; Song & Ermon, 2019; Nichol & Dhariwal, 2021), text-to-image generation (Kim et al., 2022; Ramesh et al., 2022), and realistic video creation (Esser et al., 2023; Khachatryan et al., 2023; Ceylan et al., 2023). Besides, DPMs are not limited to classical supervised learning tasks mentioned above. More broadly, diffusion-based RL methods have also shown huge potential in sequential decision-making problems (Wang et al., 2022b; Fu et al., 2020), facilitating many successful attempts in RL (Janner et al., 2022). For example, Ajay et al. (2022) propose Decision Diffuser (DD), which learns policies with the return-conditioned, constraint-conditioned, or skill-conditioned diffuser and achieves better performance in many offline RL tasks.

Given the initial states, prior studies usually adopt heuristic conditions to generate behaviors by either the action-participated or non-action-participated diffusion process (Kumar et al., 2020; Walke et al., 2023). The former generation strategy directly generates the state-action sequences, while the latter method first synthesizes state sequences and then generates actions with inverse dynamics or other models (Janner et al., 2022; Chi et al., 2023). However, regardless of which approach to be adopted, the condition of the diffusion model always plays a pivotal role in generating plausible sequences where inappropriate conditions will lead to sub-optimal policies (Wang et al., 2022b; Ajay et al., 2022; Lu et al., 2023). Heuristic conditions adopted by previous studies could cause several undesirable consequences because they do not fully consider temporal information, which is critical for understanding the dynamics, dependencies, and consequences of decisions over time in sequential modeling problems. Although some existing approaches (Ajay et al., 2022; Janner et al., 2022) are conditioned on *prospective* information, such as future returns, they usually neglect the *immediate* behaviors and the *historical* behaviors, which are important during long sequence

generation, especially in partially observable and highly stochastic environments. Since temporal dependencies are associated with the performance of diffusion models, a key question arises:

*How can we further dig into the potential of DPMs by considering the temporal properties of decision-making in RL?*

In this paper, we aim to identify temporal information from experiences, systematically understand the effects of temporal dependencies, and explicitly incorporate temporal conditions into the diffusion and generation processes of diffusion models. Specifically, we identify three distinct classes: historically-conditional, immediately-conditional, and prospectively-conditional sequence generation. Any arbitrary combination of these three conditions can be integrated, which we refer to as interchangeably-conditional sequence generation. Then, we provide a unified discussion of temporal conditions about their respective advantages, disadvantages, and connections to existing works, including potential implementation approaches and corresponding experimental results. Inspired by the above discoveries, we propose a generic temporally-conditional diffusion model and observe that this Temporally-Composable Diffuser (TCD), with the diffusion model as a sequential planner and different temporal conditions as guidance, can capture the sequential distribution information and generate conditional planning trajectories. We adopt classifier-free training, where the interchangeably-temporal conditions are composed with samples together during reverse denoising process, and perform generation by considering the interactive history, statistical current action rewards, and remaining available returns. Additionally, apart from the above-mentioned temporal condition types, we also draw inspiration from recent works (Chen et al., 2021; Bellemare et al., 2017; Koenker & Hallock, 2001; Andrychowicz et al., 2017) , and incorporate them into our proposed TCD method to extend the attention of historical sequence length, perform better estimation on out-of-distribution actions, obtain radical and conservative reward estimation, and capture more useful feature information. The sufficient experiments confirm that TCD can perform better than other baselines in various offline RL tasks, coinciding with our motivations and findings.

In summary, our main contributions are four-fold:

- We rethink the temporal dependencies of context sequences in diffusion models and find that current diffusion-based models with heuristic conditions can not fully dig into the potential of diffusion models and lead to sub-optimal performance.
- We propose the Temporally-Composable Diffuser (TCD), which can capture the temporal dependencies of sequences when performing sequential generation. Furthermore, we provide a comprehensive discussion of the effects of temporal conditions, which reveals potential improvements and helps new algorithms discovery.
- Inspired by the discussion of temporal conditions, we also consider incorporating other techniques, such as transformer-backbone, distributional RL, quantile regression, and experience replay, into our method and provide some new variants with temporal condition guidance in Appendix D.3.
- Finally, we conduct extensive experiments and discussions in Section 6 to investigate the applicability of temporal conditions. The results show that our method can surpass or match the best performance compared with other baselines.

## 2 RELATED WORK

**Offline RL.** Offline RL aims to learn the optimal policy from previously collected datasets without extra interaction (Levine et al., 2020; Kostrikov et al., 2021; Kumar et al., 2020; Ghosh et al., 2022; An et al., 2021; Ross et al., 2011; An et al., 2021). Although offline RL technologies make it possible to avoid expensive and risky data collection processes, in practice, the distribution shift between the learned policy and the data-collected policy and the overestimation of out-of-distribution (OOD) actions pose difficulties for improving performance (Kumar et al., 2020; Fujimoto & Gu, 2021; Levine et al., 2020; Rezaeifar et al., 2022). In order to solve these issues, recent works can be roughly divided into two categories. Model-free offline RL methods apply constraints on learned policy and value function to prevent inaccurate estimation of unseen actions or enhance the robustness of OOD actions by introducing uncertainty quantification (Xie et al., 2021; Kostrikov et al., 2021; Kumar et al., 2019). Model-based RL approaches propose to learn the optimal policy through planning or RL algorithms based on synthetic experiences generated from the learned dynamics (Kidambi et al., 2020; Rigter et al., 2022; Rafailov et al., 2021).

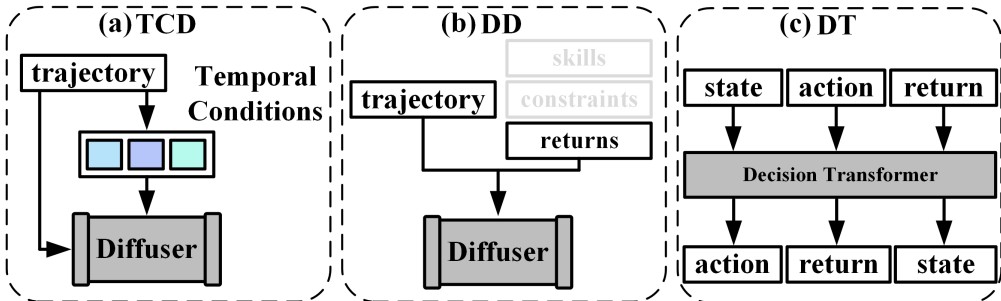

Figure 1: The comparison of TCD and the other representative baselines, such as DD and DT.

**Transformers in RL.** Recent works show huge potential in RL by casting the decision-making as a sequence generation problem (Chen et al., 2021; Janner et al., 2021; Zhang et al., 2023; Furuta et al., 2021; Wang et al., 2022a; Hu et al., 2022). For example, given the future return as prompt, the Decision Transformer (DT) orderly generates state, action, and reward tokens by considering the historical token sequence (Chen et al., 2021). Another example is the Trajectory Transformer (TT), which discretizes the state, action, and reward tokens and generates the sequences through beam search (Janner et al., 2021). Compared with Transformer-based policies, the diffusion-based methods integrate planning and decision-making together and leverage conditions to guide the whole generative sequences directly, while the prompts of transformers work iteratively.

**Diffusion Models in RL.** Diffusion models have made big progress in image synthesis and text generation by formulating the data-generating process as an iterative denoising procedure (Sohl-Dickstein et al., 2015; Rombach et al., 2022). Recently, offline decision-making problems have been formulated from the perspective of sequential distribution modeling and conditional generation with DPMs (Janner et al., 2022; Fontanesi et al., 2019; Ajay et al., 2022; Chi et al., 2023), where high-performance policies are recovered by training on the given return-labeled trajectories datasets. This new pattern brings more flexible control in offline RL, such as goal-based planning, composable constraint combination, scalable trajectory generation, and complex skill synthesis (Ajay et al., 2022; Wang et al., 2023a). For example, Janner et al. (2022) propose to combine the learned models and the trajectory optimization methods, effectively bypassing the adversarial examples that don't exist in the environment and reaching outstanding performance under proper guidance. Ajay et al. (2022) investigate how constraints and skills can be used to train DPMs and show the potential in many RL tasks. Additionally, diffusion policy is proposed as a more expressive policy, which has been used in RL, computer vision, and natural language processing (Wang et al., 2022b).

**Sequential Modeling in RL.** In sequential modeling, temporal information remains pivotal. Various studies have been developed to capture and leverage intricate temporal dependencies. They can be roughly divided into three types. RNNs preserve temporal relationships across successive time intervals through hidden states and use historical information to perform planning and make decisions (Hafner et al., 2019; 2020). Transformers leverage attention mechanisms to process all temporal positions simultaneously, furnishing an efficient means to capture relationships across tokens (Micheli et al., 2022; Chen et al., 2021). DMs employ a diffusion process to model data distribution and generate sequences from noise (Janner et al., 2022; Ajay et al., 2022). Our method belongs to this category. Compared with diffusion-based methods, we are the first to consider temporal conditions as instructions in generation and propose a generic temporally-composable diffuser.

Compared with existing works (Find Figure 1 for synoptical comparison.), we are the first to consider temporal dependencies in generation with diffusion models. We identify three types of temporal conditions, i.e., historical condition, immediate condition, and prospective condition, and propose a generic temporally-composable diffuser to produce behaviors with high performance. Besides, we conduct extensive experiments and provide a comprehensive discussion of temporal conditions.

## 3 PRELIMINARIES

In this section, we first present the relationship between decision-making and sequence generation. Then, we review the conditional generation with diffusion models.

## 3.1 REINFORCEMENT LEARNING AS SEQUENCE GENERATION

In classical reinforcement learning, the sequential decision-making problem is formulated via the Markov Decision Process (MDP), which is defined as the tuple $\mathcal{M} = \langle \mathcal{S}, \mathcal{A}, \mathcal{T}, \mathcal{R}, \rho_0, \gamma \rangle$ where $\mathcal{S}$ and $\mathcal{A}$ represent the state and action space, respectively, $\mathcal{T} : \mathcal{S} \times \mathcal{A} \to \Delta(\mathcal{S})$ denotes the Markovian transition probability, $\mathcal{R} : \mathcal{S} \times \mathcal{A} \times \mathcal{S} \to \mathbb{R}$ is the reward function, $\rho_0$ is the initial state distribution, and $\gamma \in [0, 1)$ is the discount factor. At each time step $t$, the agent receives a state $s_t$ from the environment and produces an action $a_t$ with a stochastic or deterministic policy $\pi$. Then a reward $r_t = r(s_t, a_t)$ from the environment serves as the feedback to the executed action of the agent. After the interactive interaction with the environment in a whole episode, we will obtain the state, action, and reward sequence $\tau = \{s_t, a_t, r_t\}_{t \geq 0}$. In RL, our goal is to find a policy $\pi$ that can maximize the discounted return $\mathbb{E}_{\rho_0, \pi}[\sum_{t=0}^{\infty} \gamma^t r(s_t, a_t)]$ (Sun et al., 2022; 2023).

Each trajectory $\tau$ can be regarded as a data point sampled from trajectory distribution according to certain policy $\pi$. Then we can use diffusion models to learn the data distribution $q(\tau) = \int q(\tau^{0:K}) d\tau^{1:K}$ with a predefined forward diffusion process $q(\tau^k | \tau^{k-1}) = \mathcal{N}(\tau^k; \sqrt{\alpha_k} \tau^{k-1}, (1 - \alpha_k) \boldsymbol{I})$ and a trainable generative process $p_\theta(\tau^{k-1} | \tau^k) = \mathcal{N}(\tau^{k-1}; \mu_\theta(\tau^k, k), \Sigma_k)$, where $\alpha_{0:K}$ is preassigned, $\mu_\theta(\tau^k) = \frac{1}{\sqrt{\alpha_k}}(\tau_k - \frac{\beta_k}{\sqrt{1 - \bar{\alpha}_k}} \epsilon_\theta(\tau^k, k))$, $\Sigma_k = \frac{1 - \bar{\alpha}_{k-1}}{1 - \bar{\alpha}_k} \beta_k \boldsymbol{I}$, and $\alpha_k + \beta_k = 1$. Finally, the decision-making problem can be formulated as a sequential generation problem by learning a noising model $\epsilon_\theta(\tau^k, k)$ of the trajectory denoising process to capture the trajectory distribution and generate the offline datasets samples when a start state $s_t$ is given (Sohl-Dickstein et al., 2015; Ho et al., 2020). The simplified objective for training the diffusion model is defined by

$$\mathcal{L}(\theta) = \mathbb{E}_{k \sim U(1,2,...,K), \epsilon \sim \mathcal{N}(0,\boldsymbol{I}), \tau^0 \sim D}[||\epsilon - \epsilon_\theta(\tau^k, k)||_2^2],$$

where $k$ is the diffusion time step, $U$ denotes uniform distribution, $\epsilon$ denotes the multivariant Gaussian noise, $\tau^0 = \tau$ is sampled from the replay buffer $D$, and $\theta$ is the parameters of model $\epsilon$.

## 3.2 CONDITIONAL DIFFUSION PROBABILISTIC MODELS

There are two methods, classifier-guided and classifier-free, to train conditional diffusion models $p(\tau^{k-1} | \tau^k, \mathcal{C})$, i.e., generating data $\tau^{k-1}$ under perturbed variable $\tau^k$ and condition $\mathcal{C}$ (Liu et al., 2023). The former method enables us to first train an unconditional diffusion model, which can be used to perform conditional generation under the gradient guidance of an additional classifier. For the stochastic sampling process, such as DDPM (Ho et al., 2020), $p_{\theta,\phi}(\tau^{k-1} | \tau^k, \mathcal{C}) \propto p_\theta(\tau^{k-1} | \tau^k) p_\phi(\mathcal{C} | \tau^k)$ indicates that the classifier guidance information is $p_\phi(\mathcal{C} | \tau^k)$, where the condition $\mathcal{C}$ should be the label of data $\tau^k$ and $\phi$ is the parameters. Applying Taylor expansion on $log\, p_\phi(\mathcal{C} | \tau^k)$ at $\mu_\theta$, we can obtain the perturbed noise $\Sigma \cdot \nabla log\, p_\phi(\mathcal{C} | \tau)$ that is added during the generation process. Then we have $p(\tau^{k-1} | \tau^k, \mathcal{C}) = \mathcal{N}(\mu_\theta + \Sigma_k \cdot \nabla log\, p_\phi(\mathcal{C} | \tau), \Sigma_k)$. For the deterministic sampling process, such as DDIM (Song et al., 2020), the score function of joint distribution $p(\tau^k, \mathcal{C})$ is defined by $\nabla_{\tau^k} log\, (p_\theta(\tau^k, k) p_\phi(\mathcal{C} | \tau^k)) = -\frac{1}{\sqrt{1 - \bar{\alpha}_k}} \epsilon_\theta(\tau^k) + \nabla_{\tau^k} log\, p_\phi(\mathcal{C} | \tau^k)$. The perturbed noise is $\epsilon_\theta(\tau^k, k) - \omega \sqrt{1 - \bar{\alpha}_k} \nabla_{\tau^k} log\, p_\phi(\mathcal{C} | \tau^k)$, where $\omega$ is the guidance scale.

The classifier-free method builds the correlation between the samples and conditions in the training phase by learning unconditional and conditional noise $\epsilon_\theta(\tau^k, \emptyset, k)$ and $\epsilon_\theta(\tau^k, \mathcal{C}, k)$, where we usually choose zero vector as $\emptyset$ (Ajay et al., 2022). Then the perturbed noise at each diffusion time step is calculated by $\epsilon_\theta(\tau^k, \emptyset, k) + \omega(\epsilon_\theta(\tau^k, \mathcal{C}, k) - \epsilon_\theta(\tau^k, \emptyset, k))$. In this paper, we adopt the classifier-free guidance because it can usually bring more controllable generation and higher performance.

## 4 RETHINK THE TEMPORAL DEPENDENCIES IN SEQUENTIAL GENERATION

In this section, we rethink the temporal dependencies of sequences when regarding decision-making as the sequential generation with diffusion models and discuss the limitations of existing diffusers.

**What are the particularities when regarding decision-making as sequential generation?** Classical generation tasks, such as image synthesization, possess the one-step property, where each picture does not have temporal dependencies. But in decision-making tasks, each sequence contains temporal dependencies among the context transitions in it. Roughly assimilating trajectory with image neglects the characteristics of multi-step interaction and temporal correlation of RL. Thus, in

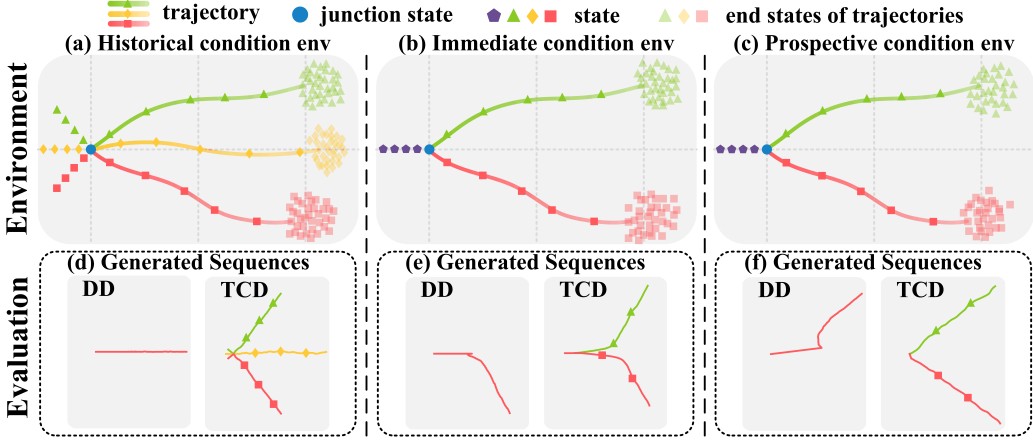

Figure 2: Temporal dependencies in RL. To intuitively observe the capacity of leveraging temporal conditions as guidance for decision-making, three types of RL tasks, **a**, **b**, and **c**, are designed for comparison. The below figures, **d**, **e**, and **f**, are the corresponding evaluations, where the results show that TCD can distinguish various sequences through temporal information and generate diverse sequences with temporal conditions guidance.

order to learn policies from the sequential data, we must extract temporal information and utilize it to guide the generation process of diffusion models.

**How do temporal conditions affect the sequential generation?** Temporal condition is important and meaningful to make decisions. For example, we will refrain from performing the salt-addition operation if we recall that we have already added salt. To visualize the effects of temporal dependencies in the sequential generation, we abstract real tasks and introduce the temporal condition env shown in Figure 2. The discrete points with different colors represent the states that contain the coordinate information. The solid lines denote state trajectories. Figures 2 (a), (b), and (c) are the datasets, and Figures 2 (d), (e), and (f) are the corresponding sequences generated by DD and our method TCD. The experimental results show that our method (TCD) can distinguish various sequences with temporal conditions, while DD without temporal conditions can not generate satisfactory sequences. Refer to Appendix C.1 and D.1 for more details. Furthermore, our method can distinguish better decisions in more complex environments and use temporal conditions to guide the diffusion models to recover better decisions by sequential generation. DD, however, can not achieve that and thus reach lower performance than our method (Refer to Section 6 for more comparison.).

## 5 TEMPORALLY-COMPOSABLE DIFFUSER

In this section, we introduce the Temporally-Composable Diffuser (TCD), as shown in Figure 3, which contains three types of temporal conditions generally denoted by $\mathcal{C}_{TCG}$, and provide a unified discussion of temporal conditions. Following previous works (Janner et al., 2022; Ajay et al., 2022), we use inverse dynamics to produce actions based on the state sequence generated by the diffusion model because 1) The state is usually continuous in RL, but the mode of action is diverse, such as discrete and continuous. 2) The action is usually high-frequency and less smooth, such as the joint torques, making it hard to model and predict the action sequence. Due to the discrepancy across tasks, various trajectory length makes it hard to train diffusion models. So, the trajectories are split into equaling sequences with $T$ time steps. We use the hat symbol to denote the generative sequence in the following parts and the variables without the hat symbol to represent the training data.

We use the diffusion model during training to capture the joint distribution of $\mathcal{C}_{TCG}$ and the state sequences. During generation, we search the temporal conditions from previously collected experiences and interactive sequences. The universal training objective and generation process are

$$\mathcal{L}(\theta) = \mathbb{E}_{k \sim U(1,2,...,K),\epsilon \sim \mathcal{N}(0,\boldsymbol{I}),\tau^0 \sim D}[||\epsilon - \epsilon_\theta(\tau_s^k, \mathcal{C}_{TCG}, k)||_2^2], \tag{1}$$

$$\tau_s^{k-1} = \frac{\sqrt{\bar{\alpha}_{k-1}}\beta_k}{1 - \bar{\alpha}_k} \cdot \bar{\tau}_s + \frac{\sqrt{\alpha}_k(1 - \bar{\alpha}_{k-1})}{1 - \bar{\alpha}_k}\tau_s^k + |\Sigma_k|\boldsymbol{z}, \tag{2}$$

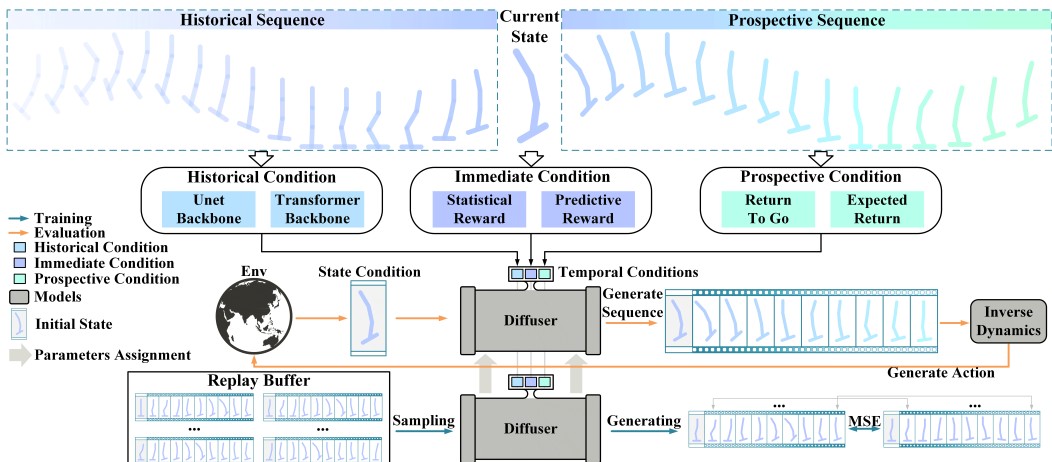

Figure 3: The overall architecture of TCD. For the whole sequence shown on the top, we select one state as the current state. Then, we can obtain the historical sequence and prospective sequence. During the training stage, we extract temporal information from the two separated sequences and the reward of the current state to train the diffusion model. During the evaluation stage, we adopt the temporal conditions that are obtained from the top-$Y$ trajectories to guide the generation process.

where $\mathcal{C}_{TCG}$ represents the mixture of prospective condition $\mathcal{C}_{PC}$, historical condition $\mathcal{C}_{HC}$, and immediate condition $\mathcal{C}_{IC}$. $\bar{\epsilon} = \epsilon_\theta(\tau_s^k, \emptyset, k) + \omega(\epsilon_\theta(\tau_s^k, \mathcal{C}, k) - \epsilon_\theta(\tau_s^k, \emptyset, k))$, $\bar{\tau}_s = \frac{\tau_s^k - \sqrt{1-\bar{\alpha}_k}\bar{\epsilon}}{\sqrt{\bar{\alpha}_k}}$, $|\Sigma_k| = \frac{1-\bar{\alpha}_{k-1}}{1-\bar{\alpha}_k}\beta_k$, and $z \sim \mathcal{N}(\mathbf{0}, \boldsymbol{I})$. Next, we introduce these temporal conditions successively.

**Prospective Condition $\mathcal{C}_{PC}$.** For each sequence $\{\hat{s}_t, \hat{a}_t, \hat{r}_t\}_{t:t+T-1}$ to be waiting for generation, the prospective condition information can be provided as guidance, such as expected discounted return of state value or state-action value, RTG, and target goal state. Compared with previous diffusion-based methods that estimate the action value function Q or state value function V, the advantage of RTG is that modeling the RTG bypasses inaccurate estimation of value on OOD samples. Besides, RTG relates the remaining available time step with the historical best performance in one episode, which can not be reflected by the Q or V function.

In this paper, we adopt the RTG, which indicates the future desired returns when preestablishing the initial returns. In the generation process, we select the maximal return value in the datasets as the initial returns. Statistically, we first calculate the episode returns from the replay buffer and then obtain the maximal return value $\mathcal{T}_{max}$ and minimum return value $\mathcal{T}_{min}$. After that, the RTG information from state $s_t$ is normalized by $\mathcal{J}_t = \frac{\mathcal{J}_t - \mathcal{T}_{min}}{\mathcal{T}_{max} - \mathcal{T}_{min}}$. Finally, we use $[\mathcal{J}_t, t]$ as the condition during training, i.e., $\mathcal{C}_{PC} = [\mathcal{J}_t, t]$. Furthermore, we find that when we slightly increase the return condition bigger than the return condition during training, the performance will be better in several tasks. More discussion can be found in Section 6.4. Apart from the conditions $\mathcal{C}_{PC}$ that are implicitly added into the model training, similar to the processing way in Diffuser, where they apply the $s_t$ condition by replacing the denoised sequence $\hat{s}_{t:t+T-1}$ with $\{s_t, \hat{s}_{t+1:t+T-1}\}$, we also explicitly leverage this type of condition in our method.

**Historical Condition $\mathcal{C}_{HC}$.** Classical diffusion model structure utilizes the U-net backbone and one-dimensional convolution to process sequence data, so the most straightforward method to consider historical information is conditioning on preceding experiences. Specifically, based on the U-net backbone, we add the historical information by replacing the incipient generative state segments $\{\hat{s}_t, ..., \hat{s}_{t+T_{HC}-1}\}$ of sequences with $\mathcal{C}_{HC} = [\mathcal{H}] = \{s_t, ..., s_{t+T_{HC}-1}\}$ at each generative step. Although previous methods, like DD, mentioned that they use historical information in their algorithm, they mainly focus on combining constraints and skills as conditions during generation and do not analyze the effect of historical information in detail. It remains unclear whether depending on historical context can benefit the performance and how to implement it properly. In contrast, we mainly focus on temporal conditions and use them to instruct the diffusion models. We provide a systematical study of temporal conditions, detailed implementation, and exhaustive experiments (See Section 6 for detailed experimental comparison.).

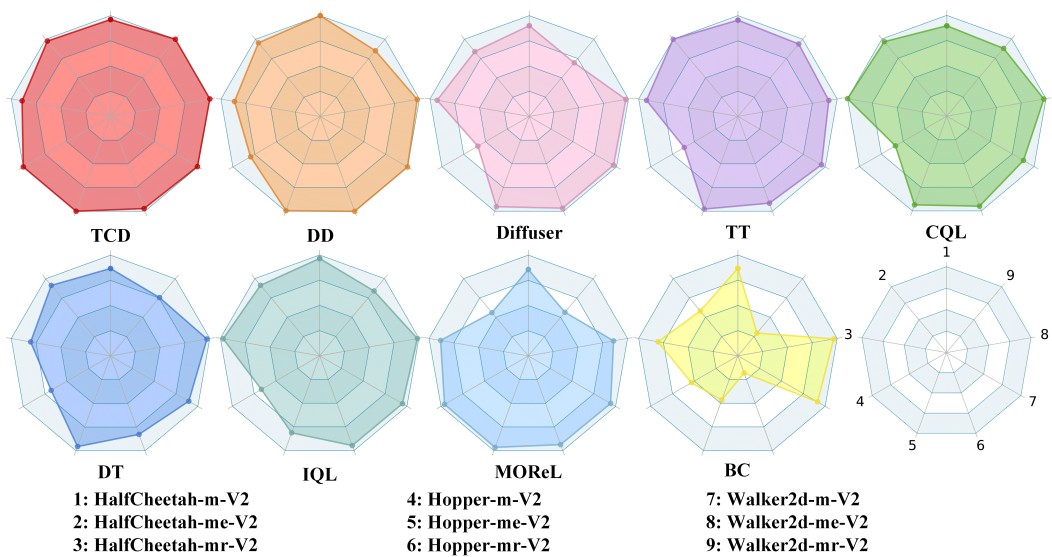

Figure 4: Offline RL algorithms comparison on D4RL Gym-MuJoCo datasets under successful comparison. The area of the polygon represents holistic performance. Each vertex of the polygon denotes the normalized performance across all methods.

**Immediate Condition** $\mathcal{C}_{IC}$. Though we can use the diffusion model to plan a long-term sequence, only the first two states, $s_t$ and $\hat{s}_{t+1}$, are adopted to produce actions with inverse dynamics $a_t = f_{inv}(s_t, \hat{s}_{t+1})$. Consequently, the direct influencing factor in obtaining rewards from the environment is the quality of the generated states $\hat{s}_{t+1}$. This enlightens us that we should pay more attention to the current generative state $\hat{s}_{t+1}$, which is the meaning of immediate condition $\mathcal{C}_{IC}$.

The immediate condition $\mathcal{C}_{IC}$ works through two stages. During the training stage, the first action reward $r_t$ and the corresponding sequence $\{s_t\}_{t:t+T-1}$ are bound together for training. Then, in the evaluation stage, we select the $Y$ trajectories with top-$Y$ returns and extract the reward sequences $\tau_r$ as the immediate condition. For each time step $t$ in the evaluation, we use the $\tau_r[t]$ that associates $s_t$ and $a_t$ to instruct generation. Thus, $\mathcal{C}_{IC} = [\tau_r[t]]$ is the immediate condition.

## 6 EXPERIMENTS

In this section, we investigate the effects of different temporal conditions on a variety of different decision-making tasks (Fu et al., 2020; Fujimoto et al., 2019; Rajeswaran et al., 2017). We first introduce environmental settings in Section 6.1. Then, in Section 6.2, we report the performance of TCD on various tasks. Next, we provide the discussion of temporal conditions in Section 6.3 and Appendix E. Finally, we conduct parameter sensitivity analysis in Section 6.4.

### 6.1 ENVIRONMENT SETTINGS

**Environments.** We select three domains with different tasks for evaluation. From Gym-MuJoCo, we select several offline datasets (HalfCheetah, Hopper, Walker2d) with different difficulty (-random (**-r**), -medium (**-m**), -expert (**-e**), -medium-replay (**-mr**), -medium-expert (**-me**), and -full-replay (**-fr**)) (Fu et al., 2020). In Maze2D and Hand Manipulation, we also select several tasks under sparse and dense reward functions (Fujimoto et al., 2019). See Appendix C.3 and C.4 for more description.

**Baselines.** We compare our method with recent representative offline RL baselines (Figure 4), including model-based methods MOReL (Kidambi et al., 2020), model-free methods CQL (Kumar et al., 2020), IQL (Kostrikov et al., 2021), BC, and sequential modeling methods DT (Chen et al., 2021), TT (Janner et al., 2021), Diffuser (Janner et al., 2022), and DD (Ajay et al., 2022).

**Metrics.** For all methods considered, we report the performance under different offline RL datasets. In order to compare the holistic capacity of all methods, the performance normalization based on all

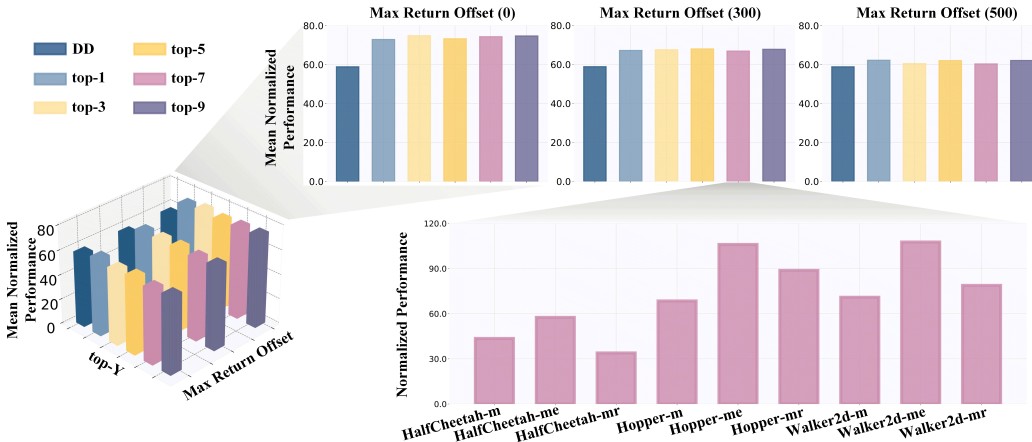

Figure 5: Parametric sensitivity of TCD on top-$Y$ and max return offset under general comparison.

methods is considered because the value of performance in different scenarios varies considerably. For example, if we have evaluation results $(E_1, E_2, E_3)$ of three methods, then the normalized performance is defined by $E'_i = \frac{\omega(E_i - min(E_1, E_2, E_3))}{max(E_1, E_2, E_3) - min(E_1, E_2, E_3)}$, where $i \in \{1, 2, 3\}$ and $\omega$ enables us to distinguish the differences of methods appropriately. In the experiments, we set $\omega = 10$. Note that we also report the normalized score on environments in Appendix D.

## 6.2 Evaluation on the Effectiveness of TCD

**Evaluation on Gym-MuJoCo.**  Results for the Gym-MuJoCo domains are shown in Figure 4 and Table 1. The results for the baselines are based on the numbers reported in (Ajay et al., 2022), (Janner et al., 2022), and (Janner et al., 2021). On the datasets generated from single policy and multi policies, TCD surpasses or matches the best prior methods in 8 of 9 environments under successful comparison, i.e., the evaluation time step is equal to the time limit of environments, and the agent is still capable of gaining more rewards. Previous diffusion-based models (i.e., Diffuser and DD) do not realize the temporal dependencies among prospective, immediate, and historical decisions. In comparison, our method guides generation with these temporal conditions and outperforms baselines by large margins in Walker2d-mr and Hopper-m.

**Evaluation on Maze2D and Hand Manipulation.**  We report the overall performance in Table 2 by comparing our method (TCD) with other baselines in Maze2D and Hand Manipulation tasks, where the average scores show that our method performs better than other baselines on these two tasks. In the Maze2D dense environment, DD performs better than TCD in the umaze and medium scenarios, but our method performs better than DD in large scenarios. The reason is that there is a tradeoff between the useful guidance and additional condition variables brought from the temporal condition. Compared with the umaze and medium scenarios, historical information is important for the agent to remember the past experience and make good decisions. While in umaze and medium scenarios, the additional condition variables bring more difficulty for the training. In the Hand Manipulation environment, all methods do not perform well on Relocate human and cloned tasks because Hand Manipulation tasks are substantially more difficult than Gym-MuJoCo because of the dataset composition, dataset volume, and high dimensionality. In the Relocate environment, the fine control needed to manipulate the 24-DoF robotic hand to complete the tasks brings much more difficulty to solving these tasks and we will devote ourselves to finding better methods for fine control with diffusion models in future work.

## 6.3 Ablation Study of Temporal Conditions

To investigate the effects of different temporal conditions, we report the performance of TCD when removing certain temporal conditions. Specifically, TCD w/o $\mathcal{C}_{HC,IC}$ denotes that we remove the historical condition $\mathcal{C}_{HC}$ and immediate condition $\mathcal{C}_{IC}$ during training and evaluation. TCD w/o $\mathcal{C}_{IC}$ denotes TCD without $\mathcal{C}_{IC}$ guidance, while TCD w/o $\mathcal{C}_{HC}$ means that we remove the historical

Table 1: Offline RL algorithms comparison on Gym-MuJoCo datasets under successful comparison.

| Dataset | Med-Expert | | | Medium | | | Med-Replay | | | score |
|---|---|---|---|---|---|---|---|---|---|---|
| Env | HalfCheetah | Hopper | Walker2d | HalfCheetah | Hopper | Walker2d | HalfCheetah | Hopper | Walker2d | |
| BC | 55.2 | 52.5 | 107.5 | 42.6 | 52.9 | 75.3 | 36.6 | 18.1 | 26.0 | 51.9 |
| MOReL | 53.3 | 108.7 | 95.6 | 42.1 | 95.4 | 77.8 | 40.2 | 93.6 | 49.8 | 72.9 |
| DT | 86.8 | 107.6 | 108.1 | 42.6 | 67.6 | 74.0 | 36.6 | 82.7 | 66.6 | 74.7 |
| Diffuser | 79.8 | 107.2 | 108.4 | 44.2 | 58.5 | 79.7 | 42.2 | 96.8 | 61.2 | 75.3 |
| IQL | 86.7 | 91.5 | 109.6 | 47.4 | 66.3 | 78.3 | 44.2 | 94.7 | 73.9 | 77.0 |
| CQL | 91.6 | 105.4 | 108.8 | 44.0 | 58.5 | 72.5 | 45.5 | 95.0 | 77.2 | 77.6 |
| TT | 95 | 110.0 | 101.9 | 46.9 | 61.1 | 79.0 | 41.9 | 91.5 | 82.6 | 78.9 |
| DD | $90.6_{\pm1.3}$ | $111.8_{\pm1.8}$ | $108.8_{\pm1.7}$ | $49.1_{\pm1.0}$ | $79.3_{\pm3.6}$ | $82.5_{\pm1.4}$ | $39.3_{\pm4.1}$ | $100_{\pm0.7}$ | $75_{\pm4.3}$ | 81.8 |
| TCD | $92.67_{\pm3.37}$ | $112.60_{\pm1.03}$ | $111.31_{\pm0.73}$ | $47.20_{\pm0.74}$ | $99.37_{\pm0.60}$ | $82.06_{\pm1.83}$ | $40.57_{\pm1.39}$ | $97.20_{\pm2.39}$ | $88.04_{\pm1.92}$ | **85.67** |

Table 2: Overall offline RL algorithms comparison on Maze2D and Hand Manipulation datasets.

| Dataset | Maze2D | | | | | | Hand Manipulation | | | | | | score |
|---|---|---|---|---|---|---|---|---|---|---|---|---|---|
| Env | sparse | | | dense | | | Pen | | | Relocate | | | |
| | umaze | medium | large | umaze | medium | large | human | expert | cloned | human | expert | cloned | |
| BC | -5.40 | 12.35 | 3.83 | -14.56 | 16.31 | 17.09 | 7.45 | 69.67 | 6.63 | 0.06 | 57.12 | 0.05 | 13.98 |
| DD | 17.16 | -3.10 | -14.19 | 83.23 | 78.17 | 22.97 | 38.23 | 8.99 | 55.12 | 0.07 | 80.31 | 0.07 | 30.59 |
| TCD | 39.99 | 28.18 | 7.68 | 29.77 | 41.44 | 75.51 | 49.88 | 35.60 | 73.30 | 0.35 | 59.64 | 0.15 | **36.80** |

Table 3: Ablation study on temporal conditions under successful comparison.

| Dataset | Med-Expert | | | Medium | | | Med-Replay | | | score |
|---|---|---|---|---|---|---|---|---|---|---|
| Env | HalfCheetah | Hopper | Walker2d | HalfCheetah | Hopper | Walker2d | HalfCheetah | Hopper | Walker2d | |
| TCD w/o $\mathcal{C}_{HC,IC}$ | $89.11_{\pm4.47}$ | $113.13_{\pm1.55}$ | $101.60_{\pm9.90}$ | $43.58_{\pm1.26}$ | $98.66_{\pm1.03}$ | $79.23_{\pm3.09}$ | $42.01_{\pm1.23}$ | $97.27_{\pm0.00}$ | $81.26_{\pm4.30}$ | 82.87 |
| TCD w/o $\mathcal{C}_{HC}$ | $91.56_{\pm2.80}$ | $113.63_{\pm1.98}$ | $109.62_{\pm0.66}$ | $44.48_{\pm0.72}$ | $98.05_{\pm2.00}$ | $78.55_{\pm3.65}$ | $40.61_{\pm1.58}$ | $99.21_{\pm1.59}$ | $89.18_{\pm2.65}$ | 84.98 |
| TCD w/o $\mathcal{C}_{IC}$ | $93.44_{\pm1.62}$ | $112.51_{\pm0.46}$ | $108.75_{\pm0.31}$ | $44.63_{\pm0.63}$ | $99.91_{\pm0.35}$ | $83.16_{\pm1.70}$ | $40.49_{\pm1.42}$ | $98.34_{\pm1.61}$ | $86.57_{\pm2.11}$ | 85.31 |
| TCD | $92.67_{\pm3.37}$ | $112.60_{\pm1.03}$ | $111.31_{\pm0.73}$ | $47.20_{\pm0.74}$ | $99.37_{\pm0.60}$ | $82.06_{\pm1.83}$ | $40.57_{\pm1.39}$ | $97.20_{\pm2.39}$ | $88.04_{\pm1.92}$ | 85.67 |

condition $\mathcal{C}_{HC}$. We conduct the experiments on Gym-MuJoCo tasks under successful comparison. As shown in Table 3, each type of temporal condition contributes positive effects in the sequential generation. Besides, $\mathcal{C}_{HC}$ leads to greater performance gain compared with $\mathcal{C}_{IC}$ when considering the performance difference between TCD w/o $\mathcal{C}_{IC}$ and TCD w/o $\mathcal{C}_{HC}$. The reason is that $\mathcal{C}_{HC}$ provides multi-step information while $\mathcal{C}_{IC}$ only contains one-step information. More extensively, we consider other explorations of the temporal conditions, including explicit and implicit temporal conditions types, and report the results in Figure 6 and Table 7. More discussion of the temporal conditions ranging from architecture backbone to training mode can be found in Appendix E.

### 6.4 PARAMETER SENSITIVITY ANALYSIS

In this experiment, we investigate several hyper-parameters that may influence the performance of temporal conditions. Specifically, we investigate the impacts of top-$Y$ and max return offset. Top-$Y$ denotes that the action-instructed reward sequence is selected from the trajectories of top-$Y$ expected return, while the max return offset represents the value that we add into the initial RTG setting during the evaluation stage. As shown in Figure 5, small $Y$, such as $Y = 1$, is good enough for generating plausible sequences, but due to poor robustness to deflected situations, small $Y$ may result in low performance in Walker2d-m. When we set a larger value of max return offset, the model may show over-optimistic to future return and weaken the effects of the current situation and historical behaviors, thus leading to performance decay. But the above phenomenon also relates to the quality of datasets where relatively increasing the value of max return offset will bring higher performance. For more discussion and experiments, please refer to Appendix D.4.

## 7 CONCLUSION

In this paper, we propose Temporally-Composable Diffuser (TCD), a generic temporally-conditional diffusion model that can extract temporal dependencies of sequences and achieve better controllable generation in sequential modeling. We identify the historical conditions, immediate conditions, and prospective conditions from sequences and provide a comprehensive discussion and comparison of different temporal conditions. We evaluate TCD on extensive experiments, including D4RL, Maze2D, and Hand Manipulation datasets, where experiments demonstrate the superiority of our method compared with other sequential modeling methods and non-sequential modeling methods.

## REPRODUCIBILITY STATEMENT

**(This section does not count towards the page limit.)**

We provide the detailed algorithm description and experimental implementation details in Appendix B. We will make our codes and pre-trained checkpoints publicly available to facilitate the replication and verification of our results upon publication.

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

APPENDIX OF PAPER "INSTRUCTED DIFFUSER WITH TEMPORAL CONDITION GUIDANCE FOR OFFLINE REINFORCEMENT LEARNING"

## A  PSEUDOCODE OF TEMPORALLY-COMPOSABLE DIFFUSER (TCD)

The pseudocode for TCD training is shown in Algorithm 1. The source code is available at https://anonymous.4open.science/r/Temporally-Composable-Diffuser-830E.

---

**Algorithm 1** Temporally-Composable Diffuser (TCD)

---

1: **Input:** Diffusion model noise prediction model $\epsilon_\theta$, Inverse dynamics model $f_\phi$
2: **Output:** $\epsilon_\theta$, $f_\phi$
3: **Requirements:** max diffusion step $K$, sequence length $L$, env time limit $t_{max}$, historical condition length $T_{HC}$, state dimension $d_s$, action dimension $d_a$, reply buffer $D$, noise schedule $\alpha_{0:K}$ and $\beta_{0:K}$
4: **Initialization:** $\theta$, $\phi$
5: // **Prepare for Training**
6: Separate the state trajectories of $D$ into state segments with length $L$
7: Normalize state segments to obey uniform distribution
8: Mark the first $T_{HC}$ states as historical condition $\mathcal{C}_{HC}$
9: Mark the action reward $r_{T_{HC}}$ of state $s_{T_{HC}}$ as immediate condition $\mathcal{C}_{IC}$
10: Find the min and max trajectory return $\mathcal{T}_{min}, \mathcal{T}_{max}$ from $D$
11: Mark the trajectory returns normalized with $\mathcal{T}_{min}, \mathcal{T}_{max}$ as $\mathcal{C}_{PC}$
12: Construct the temporal condition $\mathcal{C}_{TCG}$ by wrapping $\mathcal{C}_{HC}$, $\mathcal{C}_{IC}$, and $\mathcal{C}_{PC}$
13: // **Training Stage**
14: **for** each train iteration **do**
15:     **for** each train step **do**
16:         Sample $b$ state sequences $\tau_s^0 = \{s_{i:i+L}\} \in \mathbb{R}^{b \times L \times d_s}$, RTGs $\tau_{RTG} = \{\mathcal{T}\} \in \mathbb{R}^{b \times 1}$, action rewards $r_a = \{\tau_r[i]\} \in \mathbb{R}^{b \times 1}$, and time steps $\tau_t = \{i\} \in \mathbb{R}^{b \times 1}$ from $D$
17:         Sample diffusion time step $k \sim \text{Uniform}(K)$
18:         Obtain $\tau_s^k$ by adding noise to $\tau_s^0$
19:         Sample gaussian noise $\epsilon \sim \mathcal{N}(0, \boldsymbol{I}), \epsilon \in \mathbb{R}^{b \times L \times d_s}$
20:         Train $\epsilon_\theta$ with $\mathcal{C}_{TCG} = [r_a, \tau_{RTG}, \tau_t]$ according to Equation 1
21:         Train $f_\phi$ with $\mathbb{E}_{\tau_s^0, \tau_a^0}[||\tau_a^0 - f_\phi(\tau_s^0[:, 0:2, :])||^2]$
22:     **end for**
23:     Save model periodically
24: **end for**
25: // **Prepare for Evaluation**
26: Select the top-$Y$ trajectories $\{\tau_i\}_{i \in Y}$ according to the returns
27: Calculate the mean reward trajectories $\tau_r$ by average on $\{\tau_{r,i}\}_{i \in Y}$
28: $t = 0, \mathcal{T} = \mathcal{T}_{max}$
29: // **Evaluation Stage**
30: **for** each evaluation step **do**
31:     Receive state $s_t$ from the environment
32:     Let $k = K$
33:     Sample $\hat{\tau}_s^k \in \mathbb{R}^{1 \times L \times d_s}$ from normal distribution $\mathcal{N}(0, \boldsymbol{I})$
34:     // Replace the first $T_{HC}$ items of $\hat{\tau}_s^k$ with $s_t$
35:     **if** $t < T_{HC}$ **then**
36:         Perform sequence padding with zero states and $s_t$
37:     **else**
38:         Perform sequence padding with historical states and $s_t$
39:     **end if**
40:     Construct $\mathcal{C}_{TCG} = [\tau_r[t], \mathcal{T}, t]$
41:     Generate sequences $\hat{\tau}$ according to Equation 2
42:     Observe reward $r$ from the environment
43:     $t = t + 1, \mathcal{T} = \mathcal{T} - r$
44: **end for**

---

As shown in lines $6-12$, we first process the sequences in the replay buffer to obtain the temporal conditions. Then, during the training stage (lines $14-23$), we use the diffusion model to model the joint distribution between the sequences $\tau_s$ and the temporal conditions $\mathcal{C}_{TCG}$. After getting the well-trained diffusion model, we select the top-$Y$ trajectories and extract $\mathcal{C}_{IC}$ and $\mathcal{C}_{PC}$ (lines $26-27$) from these trajectories. We construct $\mathcal{C}_{HC}$ during the evaluation. Finally, during the evaluation stage (lines $30-44$), we leverage the temporally-composable condition $\mathcal{C}_{TCG} = [\tau_r[t], \mathcal{T}, t]$ to guide the state sequence generation and obtain actions with inverse dynamics.

## B  EXPERIMENTAL DETAILS

### B.1  COMPUTATIONAL RESOURCE DESCRIPTION

Experiments are carried out on NVIDIA GeForce RTX 3090 GPUs and NVIDIA A10 GPUs. Besides, the CPU type is Intel(R) Xeon(R) Gold 6230 CPU @ 2.10GHz. Each run of the experiments spanned about 24-48 hours, depending on the hyperparameters setting and the complexity of the model and the environments.

### B.2  HYPERPARAMETERS

Table 4: The hyperparameters of TCD.

| Hyperparameter | Value |
| --- | --- |
| historical sequence length $T_{HC}$ | 5 |
| max diffusion step $K$ | 200 |
| condition guidance $\omega$ | 1.2 |
| sequence length $L$ | 100 |
| network backbone | U-net (Ronneberger et al., 2015) |
| max return offset | 0/300/500 |
| loss function | MSE |
| learning rate | $2 \cdot 10^{-4}$ |
| batch size | 32 |
| optimizer | Adam (Kingma & Ba, 2014) |
| top-$Y$ | 1/3/5/7/9 |
| $\gamma$ | 0.99 |

## C  DETAILED ENVIRONMENT DESCRIPTION

### C.1  TEMPORAL CONDITION SCENARIOS

In order to show the temporal dependencies in sequential modeling and investigate whether the previous methods with prospective discounted returns, such as DD, and whether TCD can generate satisfactory sequences, we design three types of RL environments, which are shown in Figure 2.

In the historical condition scenario shown in Figure 2 (a), the collected state sequences are generated according to three different historical state sequences. In order to show the capacity to capture historical experience and avoid the diffusion model distinguishing state sequences only from the current state, these three types of historical sequences are set to converge the same junction state (blue circle). The green triangle trajectories can only be unlocked when the historical state sequence meets the condition of state incidence at the $135°$ angle to the junction state.

In Figure 2 (b), we show the immediate condition scenario, where the historical sequences of these two datasets are the same, but the action rewards under the junction state are different. We can get the low-reward trajectory samples and high-reward trajectory samples according to the action reward of the junction state. Besides, the total returns of trajectories are the same for these two types of trajectory samples. Given the above settings, we can evaluate whether the diffusion model can focus on immediate behaviors that are most related to the current interaction step with the environment.

Table 5: Additional comparison on Gym-MuJoCo dataset.

| Dataset | Random | | | Expert | | | Full-Replay | | | score |
|---|---|---|---|---|---|---|---|---|---|---|
| Env | HalfCheetah | Hopper | Walker2d | HalfCheetah | Hopper | Walker2d | HalfCheetah | Hopper | Walker2d | |
| DD | $1.96_{\pm 0.11}$ | $5.81_{\pm 1.25}$ | $12.84_{\pm 5.66}$ | $17.40_{\pm 24.55}$ | $110.87_{\pm 2.81}$ | $103.67_{\pm 17.73}$ | $45.64_{\pm 11.35}$ | $99.60_{\pm 22.30}$ | $82.06_{\pm 5.52}$ | 53.32 |
| TCD | $3.98_{\pm 0.61}$ | $2.14_{\pm 0.82}$ | $6.50_{\pm 1.78}$ | $36.38_{\pm 28.70}$ | $112.65_{\pm 1.06}$ | $108.11_{\pm 0.94}$ | $25.65_{\pm 14.66}$ | $106.27_{\pm 0.82}$ | $96.16_{\pm 1.27}$ | **55.32** |

Table 6: Ablation study on temporal conditions under general comparison.

| Dataset | Med-Expert | | | Medium | | | Med-Replay | | | score |
|---|---|---|---|---|---|---|---|---|---|---|
| Env | HalfCheetah | Hopper | Walker2d | HalfCheetah | Hopper | Walker2d | HalfCheetah | Hopper | Walker2d | |
| TCD w/o $\mathcal{C}_{HC,IC}$ | $42.68_{\pm 24.25}$ | $92.98_{\pm 20.20}$ | $79.22_{\pm 3.08}$ | $41.81_{\pm 5.58}$ | $71.24_{\pm 18.78}$ | $79.22_{\pm 3.08}$ | $32.76_{\pm 10.07}$ | $91.01_{\pm 8.92}$ | $77.97_{\pm 10.03}$ | 66.45 |
| TCD w/o $\mathcal{C}_{HC}$ | $41.81_{\pm 29.01}$ | $109.79_{\pm 10.28}$ | $108.26_{\pm 0.46}$ | $43.61_{\pm 5.46}$ | $62.31_{\pm 18.39}$ | $68.08_{\pm 20.13}$ | $38.97_{\pm 4.89}$ | $92.69_{\pm 9.54}$ | $71.49_{\pm 19.91}$ | 72.33 |
| TCD w/o $\mathcal{C}_{IC}$ | $76.92_{\pm 25.81}$ | $111.15_{\pm 5.08}$ | $108.75_{\pm 0.31}$ | $44.63_{\pm 0.63}$ | $71.48_{\pm 16.27}$ | $77.92_{\pm 7.96}$ | $34.24_{\pm 8.85}$ | $94.15_{\pm 6.49}$ | $75.33_{\pm 14.99}$ | 77.17 |
| TCD | $74.30_{\pm 29.52}$ | $110.94_{\pm 8.99}$ | $111.31_{\pm 0.73}$ | $46.73_{\pm 1.93}$ | $74.48_{\pm 19.95}$ | $73.61_{\pm 13.26}$ | $35.29_{\pm 10.60}$ | $93.32_{\pm 8.37}$ | $78.83_{\pm 13.01}$ | **77.65** |

In the prospective condition scenario, as shown in Figure 2 (c), the rewards of the front part of the upper trajectories (green triangle) are 0, and the rewards of the latter part are 1 for each time step. In contrast, the below trajectories (red square) have the opposite reward setting, i.e., 1 for the front part and 0 for the latter part. The setting of the same returns of trajectories and different action reward distributions in trajectories makes it suitable for us to test the difference between the return-guided diffusion model and the RTG-guided diffusion model.

## C.2 GYM-MUJOCO

Apart from the environments that we introduce in Section 6.2, we also evaluate our method in other scenarios, including -random (**-r**), -expert (**-e**), and -full-replay (**-fr**). The version of all the Gym-MuJoCo environments is **-v2**.

## C.3 MAZE2D

The Maze2D is a 2D navigation task where the agent needs to reach a specific location. The goal is to find the shortest path to the evaluation location by training on a previously collected dataset. There are three difficulty settings, -umaze (**-u**), -medium (**-m**), and -large (**-l**), about Maze2D according to the size of the maze layouts. The trajectories are constructed with waypoint sequences that are generated by a PD controller (Fu et al., 2020). Apart from the above-introduced difficulty settings, there are also two reward settings: sparse reward setting and dense reward setting. Thus, we can obtain six scenarios by permutation and combination: Maze2D-sparse-u, Maze2D-sparse-m, Maze2D-sparse-l, Maze2D-dense-u, Maze2D-dense-m, and Maze2D-dense-l. The version of Maze2D is **-v1**.

## C.4 HAND MANIPULATION

Hand Manipulation (i.e., Adroit) contains several sparse-reward, high-dimensional robotic manipulation tasks where the datasets are collected under three types of situations (-human (**-h**), -expert (**-e**), and -cloned (**-c**)) (Rajeswaran et al., 2017). For example, the Relocate scenario requires the agent to pick up a ball and move it to a specific location, and the Pen is a scenario where the agent needs to get rewards by twirling a pen. The datasets with -h difficulty contain a small number of trajectories, while the datasets with -e and -c include abundant trajectories for training. Compared with Maze2D and Gym-MuJoCo, Hand Manipulation possesses a higher state dimension, harder exploration, more real expert demonstration, and more sparse reward feedback. The version is **-v1**.

# D ADDITIONAL EXPERIMENT RESULTS

## D.1 ADDITIONAL DISCUSSION OF TEMPORAL CONDITIONS SCENARIOS

The results of Temporal Conditions Scenarios are shown in the part of Figure 2, and we put the environmental description in Section C.1. We select the DD with prospective returns as a comparison. For the historical condition env, the generation process of DD needs the current state as the condi-

tion. As shown in Figure 2 (d), when we give junction state as the condition, DD can not generate the corresponding sequences because it does not consider the temporal dependencies between history and immediate state. Our method (TCD) can produce adequate trajectories that meet the historical sequence condition because of the guidance of $\mathcal{C}_{HC}$. The results of the immediate condition env are shown in Figure 2 (e). We can see that TCD can distinguish different action reward trajectories under the same trajectory returns with immediate condition $\mathcal{C}_{IC}$ guidance. However, this does not imply that our model can't generate high-reward sequences. Since our model can discern distinct sequences, it inherently possesses the capability to generate high-reward sequences. Although DD can also generate the sequence with high action reward, the single mode of generative samples reveals poor discernibility and hinders DD from applications such as diverse trajectories generation. Finally, in the prospective condition env, from Figure 2 (f), we can also see that the RTG with the remaining available time step can help the diffusion model recognize different trajectories. However, the DD can not recover all types of trajectories when the prospective returns are the same.

## D.2 ADDITIONAL EXPLORATIONS ABOUT TEMPORAL CONDITIONS

**TCD w/o $\mathcal{C}_{HC,IC}$.** Compared with previous diffusion-based methods that estimate the action value function Q or state value function V, we can obtain the RTG instruction easily from the experiences rather than suffering from inaccurate estimation on OOD samples. Besides, RTG relates the remaining available time step with the historical best performance in one episode, which can not be reflected by the Q or V function. Statistically, we first calculate the episode returns from the replay buffer and then obtain the maximal return value $\mathcal{T}_{max}$ and minimum return value $\mathcal{T}_{min}$. After that, the RTG information from state $s_t$ is normalized by $\mathcal{J}_t = \frac{\mathcal{J}_t - \mathcal{T}_{min}}{\mathcal{T}_{max} - \mathcal{T}_{min}}$. Finally, we use $[\mathcal{J}_t, t]$ as the condition during training, i.e., $\mathcal{C}_{PC} = [\mathcal{J}_t, t]$ when $PC = RTG$.

**TCD w/o $\mathcal{C}_{IC}$.** Based on the U-net backbone, we add the historical information by replacing the incipient generative state segments $\{\hat{s}_t, ..., \hat{s}_{t+T_{HC}-1}\}$ of sequences with $\mathcal{H} = \{s_t, ..., s_{t+T_{HC}-1}\}$ at each generative step. Previous methods, like DD, generate state sequence conditioning only on current states $s_t$, which may omit important information that appears in history, especially for environments with partial observability. Even for environments without partial observability, the historical conditions can be regarded as an argumentation method, and the experiments show they can still provide improvements. We also use RTG and remaining time steps as additional information during training. Then, the historical condition is defined as $\mathcal{C}_{HC} = [\mathcal{H}, \mathcal{J}_t, t]$.

**TCD w/o $\mathcal{C}_{HC}$.** During the inference stage, though we will generate a state sequence $\{\hat{s}_t\}_{t:t+T-1}$ under current state $s_t$, only state $\hat{s}_{t+1}$ is used to produce action $a_t$, which inspires us to focus on the generative quality of the current action. Thus, we adopt the first action reward $r_t$ that associates $s_t$ and $a_t$ to instruct generation, where $r_t$ is calculated from top-$Y$ trajectories in the replay buffer. Mathematically, $\mathcal{C}_{IC} = [r_t, \mathcal{J}_t, t]$ is used for training, and $\tau_r = \mathbb{E}_{\tau \sim D_{\text{top-K}}}[\tau]$ is used for evaluation, where $r_t$ and $\tau_r$ are normalized according to maximal and minimum single-step reward.

**Diffuser with Transformer-Backbone (TFD).** As for the implicit line about historical condition, we've recently seen efforts to substitute U-Net with the transformer due to the similar input-output structure. Thus, we try to replace the U-net backbone with the transformer backbone to focus on longer history information (Touvron et al., 2023). We call this method diffuser with transformer-backbone (TFD), which has lower memory consumption and shorter run time because it does not use the convolution operation.

**Diffuser with State-Reward Sequence Modeling (SRD).** For the implicitly immediate or prospective condition, we propose modeling the distribution of state sequences and the joint distribution of state sequences and the reward sequences. For the state $s_t$ of the current time step, we use SRD to generate several candidate sequences $\{s_t, \hat{r}_t, \hat{s}_{t+1}, \hat{r}_{t+1}, ..., \hat{r}_{t+L-1}\}_{n=1}^{N}$ and select the sequence with max reward $\hat{r}_t^n$ (i.e., implicitly immediate condition) or max returns $\sum_{i=1}^{t+L-1} \hat{r}_i^n$ (i.e., implicitly prospective condition).

**Diffuser with Distributional Q Estimation (DQD).** We also conduct other explorations about explicitly temporal conditions. In order to obtain better approximation by preserving the multimodality of the action-value function, we explore fitting the Q function on offline datasets with distributional RL techniques (Bellemare et al., 2017). Specifically, we separate the discounted returns into 201 bins and use two networks to predict the bin values and distributions, respectively. Then, the ex-

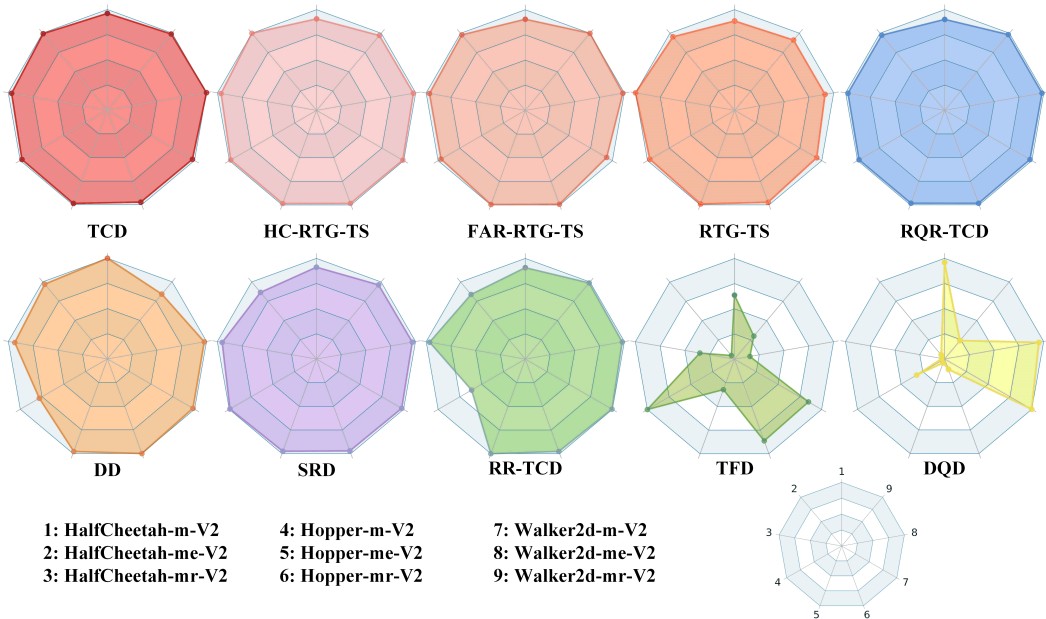

Figure 6: Evaluation of various temporal methods on D4RL Gym-MuJoCo dataset under successful comparison. The area of the polygon represents holistic performance. Each vertex of the polygon denotes the normalized performance across all methods.

Table 7: Successful comparison of the effect of various temporal conditions on diffusion model.

| Dataset | Med-Expert | | | Medium | | | Med-Replay | | | score |
|---|---|---|---|---|---|---|---|---|---|---|
| Env | HalfCheetah | Hopper | Walker2d | HalfCheetah | Hopper | Walker2d | HalfCheetah | Hopper | Walker2d | |
| DQD | $5.05_{\pm1.92}$ | $4.79_{\pm0.33}$ | $106.03_{\pm1.39}$ | $47.09_{\pm2.00}$ | $32.55_{\pm0.04}$ | $84.07_{\pm0.63}$ | $1.16_{\pm0.91}$ | $11.06_{\pm0.19}$ | $21.35_{\pm0.09}$ | 34.79 |
| TFD | $4.47_{\pm2.28}$ | $36.78_{\pm29.13}$ | $17.31_{\pm5.17}$ | $31.23_{\pm8.36}$ | $101.42_{\pm0.85}$ | $71.36_{\pm2.85}$ | $14.68_{\pm5.06}$ | $86.37_{\pm2.01}$ | $26.69_{\pm11.49}$ | 43.37 |
| RR-TCD | $78.31_{\pm28.93}$ | $114.23_{\pm1.09}$ | $108.89_{\pm0.23}$ | $44.60_{\pm0.93}$ | $62.46_{\pm13.15}$ | $83.60_{\pm1.44}$ | $40.55_{\pm1.75}$ | $97.59_{\pm1.27}$ | $88.05_{\pm2.24}$ | 79.81 |
| TCD w/o $\mathcal{C}_{HC,IC}$ | $89.11_{\pm4.47}$ | $113.13_{\pm1.55}$ | $101.60_{\pm9.90}$ | $43.58_{\pm1.26}$ | $98.66_{\pm1.03}$ | $79.23_{\pm3.09}$ | $42.01_{\pm1.23}$ | $97.27_{\pm0.00}$ | $81.26_{\pm4.30}$ | 82.87 |
| SRD | $80.79_{\pm25.59}$ | $111.40_{\pm0.62}$ | $107.72_{\pm0.72}$ | $44.81_{\pm0.77}$ | $100.58_{\pm0.02}$ | $82.29_{\pm2.11}$ | $39.95_{\pm1.33}$ | $97.14_{\pm1.52}$ | $85.76_{\pm2.74}$ | 83.38 |
| TCD w/o $\mathcal{C}_{HC}$ | $91.56_{\pm2.80}$ | $113.63_{\pm1.98}$ | $109.62_{\pm0.66}$ | $44.48_{\pm0.72}$ | $98.05_{\pm2.00}$ | $78.55_{\pm3.65}$ | $40.61_{\pm1.58}$ | $99.21_{\pm1.59}$ | $89.18_{\pm2.65}$ | 84.98 |
| RQR-TCD | $90.97_{\pm2.88}$ | $112.22_{\pm1.23}$ | $109.24_{\pm0.20}$ | $44.34_{\pm1.13}$ | $99.71_{\pm0.97}$ | $82.24_{\pm2.32}$ | $41.01_{\pm2.28}$ | $98.30_{\pm1.40}$ | $87.54_{\pm3.07}$ | 85.06 |
| TCD w/o $\mathcal{C}_{IC}$ | $93.44_{\pm1.62}$ | $112.51_{\pm0.46}$ | $108.75_{\pm0.31}$ | $44.63_{\pm0.63}$ | $99.91_{\pm0.35}$ | $83.16_{\pm1.70}$ | $40.49_{\pm1.42}$ | $98.34_{\pm1.61}$ | $86.57_{\pm2.11}$ | 85.31 |
| TCD | $92.67_{\pm3.37}$ | $112.60_{\pm1.03}$ | $111.31_{\pm0.73}$ | $47.20_{\pm0.74}$ | $99.37_{\pm0.60}$ | $82.06_{\pm1.83}$ | $40.57_{\pm1.39}$ | $97.20_{\pm2.39}$ | $88.04_{\pm1.92}$ | **85.67** |

pected bin values weighted by the probabilities of bin selection are adopted as prospective returns to guide the diffusion model.

**TCD with Reward Estimation (RR-TCD and RQR-TCD).** Directly utilizing statistical rewards from the replay buffer can not reflect the reward situation of all collected states. Thus, we hope to improve the prediction accuracy by introducing reward estimation. In practice, directly estimating reward values conditioning on states requires learning a mapping from states to rewards, which we call this method TCD with linear reward regression (RR-TCD). Another method, TCD with reward quantile regression (RQR-TCD), gives us a chance to roughly identify the radical and conservative actions through the reward distribution in each state.

In Appendix E, we review previous studies and discuss the advantages and disadvantages of the above methods, TCD, and other baselines.

## D.3 ADDITIONAL EVALUATION ON VARIOUS SCENARIOS

In this section, we report the additional evaluation on many environments, such as Pen-{h, e, c}-v1, Relocate-{h, e, c}-v1, HalfCheetah-{r, e, fr}-v2, Hopper-{r, e, fr}-v2, and Walker2d-{r, e, fr}-v2. From the results that are shown in Table 2 and Table 5, we can see that our method (TCD) achieves a 20% overall performance gain compared with DD and reaches the best mean performance in Pen and Relocate environments. In the HalfCheetah-{r, e, fr}-v2, Hopper-{r, e, fr}-v2, and Walker2d-{r, e, fr}-v2 environments, the performance of our method surpasses DD in 5 of 6 datasets with countable

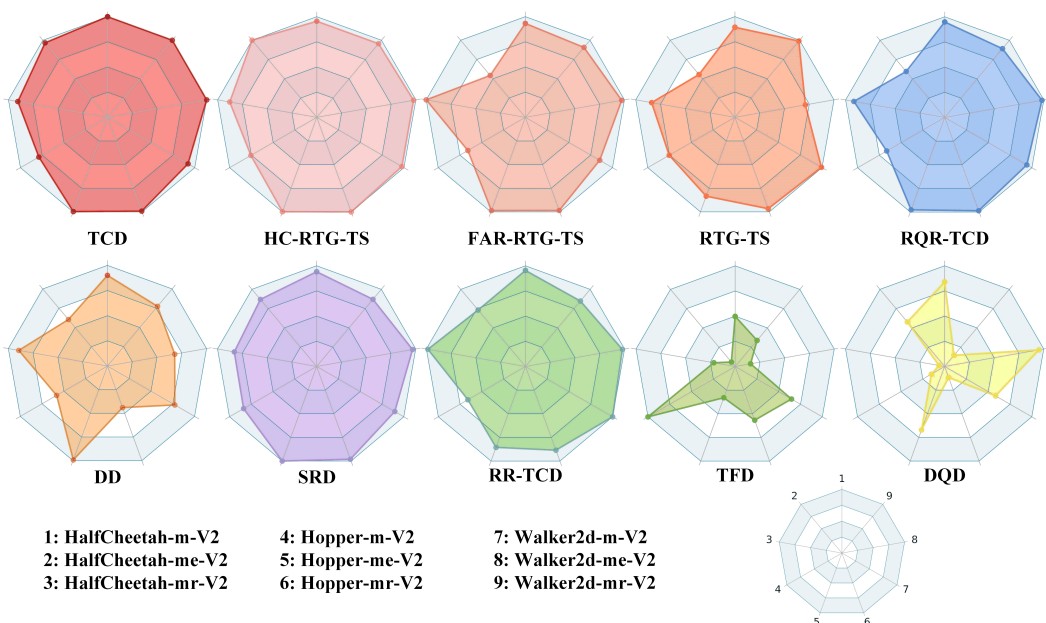

Figure 7: Evaluation of various temporal methods on D4RL Gym-MuJoCo dataset under general comparison. The area of the polygon represents holistic performance. Each vertex of the polygon denotes the normalized performance across all methods.

Table 8: General comparison of the effect of various temporal conditions on diffusion model.

| Dataset | Med-Expert | | | Medium | | | Med-Replay | | | score |
|---|---|---|---|---|---|---|---|---|---|---|
| Env | HalfCheetah | Hopper | Walker2d | HalfCheetah | Hopper | Walker2d | HalfCheetah | Hopper | Walker2d | |
| DQD | $4.43_{\pm2.49}$ | $4.79_{\pm0.33}$ | $106.03_{\pm1.39}$ | $39.28_{\pm15.12}$ | $14.44_{\pm8.13}$ | $46.68_{\pm18.69}$ | $2.22_{\pm2.45}$ | $11.06_{\pm0.19}$ | $11.51_{\pm6.50}$ | 26.77 |
| TFD | $4.47_{\pm2.28}$ | $36.78_{\pm29.13}$ | $17.31_{\pm5.17}$ | $23.27_{\pm9.48}$ | $94.47_{\pm12.93}$ | $51.55_{\pm27.52}$ | $8.38_{\pm6.24}$ | $53.43_{\pm30.23}$ | $26.69_{\pm11.49}$ | 35.62 |
| TCD w/o $\mathcal{C}_{HC,IC}$ | $42.68_{\pm24.25}$ | $92.98_{\pm20.20}$ | $79.22_{\pm3.08}$ | $41.81_{\pm5.58}$ | $71.24_{\pm18.78}$ | $79.22_{\pm3.08}$ | $32.76_{\pm10.07}$ | $91.01_{\pm8.92}$ | $77.97_{\pm10.03}$ | 66.45 |
| RR-TCD | $56.35_{\pm25.84}$ | $95.33_{\pm25.74}$ | $108.89_{\pm0.23}$ | $44.60_{\pm0.93}$ | $62.46_{\pm13.15}$ | $79.74_{\pm9.77}$ | $38.38_{\pm6.52}$ | $83.36_{\pm17.77}$ | $66.96_{\pm25.99}$ | 70.73 |
| RQR-TCD | $45.88_{\pm24.32}$ | $108.87_{\pm12.84}$ | $109.24_{\pm0.20}$ | $44.34_{\pm1.13}$ | $62.93_{\pm16.79}$ | $75.15_{\pm14.39}$ | $35.73_{\pm10.20}$ | $92.84_{\pm11.92}$ | $70.35_{\pm25.78}$ | 71.70 |
| TCD w/o $\mathcal{C}_{HC}$ | $41.81_{\pm29.01}$ | $109.79_{\pm10.28}$ | $108.26_{\pm0.46}$ | $43.61_{\pm5.46}$ | $62.31_{\pm18.39}$ | $68.08_{\pm20.13}$ | $38.97_{\pm4.89}$ | $92.69_{\pm9.54}$ | $71.49_{\pm19.91}$ | 72.33 |
| SRD | $66.85_{\pm31.52}$ | $111.40_{\pm0.62}$ | $107.72_{\pm0.72}$ | $43.96_{\pm4.64}$ | $79.17_{\pm18.36}$ | $71.50_{\pm16.40}$ | $32.38_{\pm10.09}$ | $92.33_{\pm10.20}$ | $68.61_{\pm17.20}$ | 74.88 |
| TCD w/o $\mathcal{C}_{IC}$ | $76.92_{\pm25.81}$ | $111.15_{\pm5.08}$ | $108.75_{\pm0.31}$ | $44.63_{\pm0.63}$ | $71.48_{\pm16.27}$ | $77.92_{\pm7.96}$ | $34.24_{\pm8.85}$ | $94.15_{\pm6.49}$ | $75.33_{\pm14.99}$ | 77.17 |
| TCD | $74.30_{\pm29.52}$ | $110.94_{\pm8.99}$ | $111.31_{\pm0.73}$ | $46.73_{\pm1.93}$ | $74.48_{\pm19.95}$ | $73.61_{\pm13.26}$ | $35.29_{\pm10.60}$ | $93.32_{\pm8.37}$ | $78.83_{\pm13.01}$ | **77.65** |

modality (i.e., expert and full-replay) and 6 of 9 on all datasets. The reason for poor performance in {HalfCheetah, Hopper, Walker2d} random datasets is that samples with random interaction do not possess primary modality on data distribution, thus leading to random update direction when we use the diffusion model to capture the data distribution. Finally, the diffusion model can not learn to generate good behaviors that align with certain behavior policies according to experiences.

Besides, we also realize several algorithms, such as DQD, TFD, RR-TCD, RQR-TCD, and SRD, by considering other types of temporal conditions and report the corresponding results on Gym-MuJoCo in Figure 6 and Figure 7. Although distributional Q estimation can alleviate the influence of OOD actions, the results show that directly adopting the distributional Q value as the instruction on conditional diffusion models may hurt the performance, which stimulates us to find a better way to combine the distributional techniques and diffusion models. We can lightweight the memory overhead and reduce the time consumption of training diffusion models with transformer backbone, but the results show that TFD performs poorly in most tasks because 1) the position embedding and diffusion process will introduce two different time encoding vectors, which may conflict with each other and impact the model learning. 2) The non-iterative generation mechanism. Although the generation contains multiple steps, each generation step is non-iterative, which may limit the capacity of transformers. As shown in Table 7 and Table 8, TFD (diffuser with transformer backbone) performs poorly in most tasks, where the only difference is the Hopper-m task. The reason is that the time encoding vectors may associate with the data distribution and exactly make a positive effect on the Hopper-m task. Considering the methods, RR-TCD, RQR-TCD, and SRD, that estimate the action rewards, we find that modeling the distribution of reward sequence (SRD) is better than

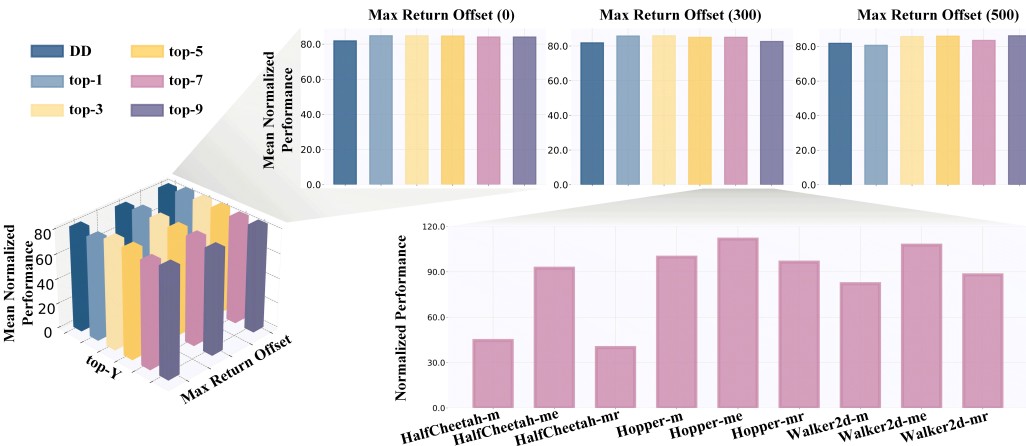

Figure 8: Parametric sensitivity about top-$Y$ and max return offset under successful comparison.

Table 9: The effects of historical sequence length $L$ under successful comparison.

| Dataset | Med-Expert | | | Medium | | | Med-Replay | | | score |
|---------|-----------|--------|---------|-------------|--------|----------|------------|--------|----------|-------|
| Env | HalfCheetah | Hopper | Walker2d | HalfCheetah | Hopper | Walker2d | HalfCheetah | Hopper | Walker2d | |
| TCD ($L5$) | $92.67_{\pm3.37}$ | $112.60_{\pm1.03}$ | $111.31_{\pm0.73}$ | $47.20_{\pm0.74}$ | $99.37_{\pm0.60}$ | $82.06_{\pm1.83}$ | $40.57_{\pm1.39}$ | $97.20_{\pm2.39}$ | $88.04_{\pm1.92}$ | **85.67** |
| TCD ($L10$) | $85.74_{\pm13.68}$ | $110.71_{\pm0.50}$ | $109.06_{\pm0.59}$ | $44.50_{\pm0.92}$ | $99.60_{\pm1.03}$ | $82.53_{\pm1.32}$ | $39.24_{\pm1.57}$ | $97.17_{\pm1.56}$ | $85.06_{\pm5.10}$ | 83.73 |

learning a mapping from state space to reward space (RR-TCD). Reward quantile regression (RQR-TCD) performs better than RR-TCD and SRD because quantile regression is insensitive to certain extreme reward values that refer to radical actions and conservative actions. Thus, more likely, we can recover the behavior policies and reach better score conditioning on the median reward values.

In Table 2, we do not include the comparison with Diffuser (Janner et al., 2022) because the reported results of Diffuser are based on goal state guidance. However, our method does not leverage the goal state for planning. Directly comparing our method with Diffuser will lose fairness. As introduced in Section 5, the goal state $s_g$ can be regarded as a type of prospective condition. We test TCD and Diffuser with and without goal state ($s_g$) in Maze2D tasks, with results in Table 10. The results show that our method surpasses Diffuser when given $s_g$. Furthermore, TCD also performs significantly better than Diffuser without $s_g$, which illustrates the importance of temporal information.

### D.4 ADDITIONAL EXPERIMENT RESULTS ABOUT PARAMETER SENSITIVITY

We report the additional experiments of parameter sensitivity from two dimensions, i.e., top-$Y$ and max return offset, in Figure 8, where the results show that our method performs better than DD in most settings of hyperparameters. Besides, we also conduct experiments about parameter sensitivity on max return offset and historical sequence length.

**Performance with Max Return Offset.** During the evaluation stage, we add max return offset to the initial RTG, where the bigger values of max return offset denote more optimism about future returns, and smaller values of max return offset indicate more pessimism about available returns. When the condition aligns well with the training data, we can appropriately increase the condition, thereby encouraging the model to discover better decision sequences from historical experiences. In several scenarios, we see the corresponding phenomena in Figure 9. The results show that we can obtain higher performance by slightly increasing the initial RTG, which inspires us to investigate adaptive methods for selecting the max return offset.

**Historical Sequence Length $L$.** We first probe the impacts of historical sequence length $L$, which represents how long the previous sequence is considered for generation when we adopt the U-net backbone. The results are shown in Table 9, where we can see that a relatively longer length of historical sequence can provide further improvements in Hopper-m and Walker2d-m. In most other scenarios, the longer historical sequence ($L = 10$) provides negative improvements because

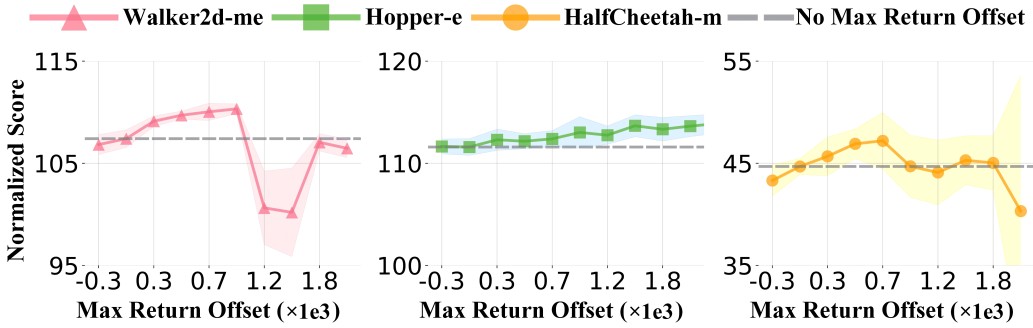

Figure 9: The performance on max return offset. The results show that when the condition aligns well with the training data, and we slightly increase the return condition bigger than the return condition during training, then the performance will be better.

Table 10: The comparison between TCD with and without $s_g$, Diffuser with and without $s_g$ in Maze2D tasks. The "w" denotes "with," and "w/o" represents "without." The results show our method can achieve better performance than Diffuser with and without $s_g$ guidance.

| Dataset | TCD w $s_g$ | Diffuser w $s_g$ | TCD w/o $s_g$ | Diffuser w/o $s_g$ |
|---|---|---|---|---|
| Maze2D-large | $146.4_{\pm34.2}$ | $123.0_{\pm6.4}$ | $7.7_{\pm14.6}$ | $-2.5_{\pm0.0}$ |
| Maze2D-medium | $132.9_{\pm34.2}$ | $121.5_{\pm2.7}$ | $28.2_{\pm5.3}$ | $-4.9_{\pm0.0}$ |
| Maze2D-umaze | $128.1_{\pm21.7}$ | $113.9_{\pm3.1}$ | $40.0_{\pm39.6}$ | $14.7_{\pm0.3}$ |

the historical sequence may introduce useful information and extra noise concurrently, where the useless noise makes it hard for learning control.

## E   MORE DISCUSSION ABOUT TEMPORAL CONDITIONS

**Prospectively-Conditional Sequence Generation.** For each $\{\hat{s}_t, \hat{a}_t, \hat{r}_t\}_{t:t+T-1}$ sequence to be waiting for generation, the prospective condition information can be provided as guidance, such as expected discounted return of state value, state-action value, RTG, and target goal state. For example, Diffuser (Janner et al., 2022) trains a Q value function separately and adds the gradient with respect to state action pairs to guide the generation, where the stochastic sampling process is $\tau^{k-1} \sim \mathcal{N}(\mu_\theta + \alpha\Sigma^k\nabla\mathcal{J}(\mu_\theta), \Sigma^k)$, $\alpha$ is the gradient scale, and $\mathcal{J} = \sum_t r(s_t, a_t)$. Limited by the restricted experiences and overestimation problem, direct Q value estimation can not provide a better approximation, while distributional RL technologies may be useful to capture the multimodal Q distribution, which leads to more stable learning (Bellemare et al., 2017; Tian et al., 2023).

The RTG, as another prospective condition form, indicates the future desired returns when pre-establishing the initial returns, which is different from the Q value function because we obtain the Q value function through temporal difference (TD) learning while the initial returns are defined according to prior knowledge of environments. In addition to the prospective conditions, goal state (GS) or goal feature (GF) can also be used to guide generation. Similar to the processing way in Diffuser, where they apply conditions of $s_t$ and $s_g$ by replacing the noisy sequence $\{\hat{s}_t\}_{t:t+T-1}$ with $\{s_t, \{\hat{s}_{t+1}\}_{t+1:t+T-2}, s_g\}$, we can also substitute the final generative state $\hat{s}_{t+T-1}$ with the goal state $s_g$ for planning. Furthermore, the goal state $s_g$ can also be embedded in the latent space, so we can use $f_s$ as a general function that represents feature mapping or identity function, i.e., $s_g = f_s^{-1}(g)$ and $g = f_s(s)$.

Most previous studies adopt prospective conditions, and the general objective function is defined as

$$\mathcal{L}(\theta) = \mathbb{E}_{k\sim U(1,2,...,T_d),\epsilon\sim\mathcal{N}(0,\boldsymbol{I}),\tau^0\sim D}[||\epsilon - \epsilon_\theta(f_\tau(\tau^k), \mathcal{C}_{PC}, k)||_2^2],$$

where $\mathcal{C}_{PC} \in \{Q, RTG, GS, GF\}$ and $f_\tau$ is a sequence processing function, which can represent state sequence, state-action sequence, state-reward sequence, or the state-action-reward sequence.

**Historically-Conditional Sequence Generation.** When we regard the decision-making problem as a sequence modeling problem, generating sequences based on $s_t$ is similar to long-term series forecasting, which motivates us to add historical information into sequence generation or preprocess the sequence data, such as firstly extracting trend variables and seasonal variables (Wu et al., 2021; Wang et al., 2023b). Classical diffusion model structure utilizes the U-net backbone and one-dimensional convolution to process sequence data, so the most straightforward method to consider historical information is conditioning on preceding experiences. From another view, the U-net backbone disregards the temporal information between consecutive transitions in a sequence, treating them as a single entity. In this context, if the sequence is considered as an image, the generation process with the U-net backbone can be likened to image inpainting.

In addition to the aforementioned methods that explicitly consider historical sequences, we can also implicitly take into account historical information. Transformers employ a novel self-attention mechanism that captures long-range dependencies and global context more effectively, leading to the widespread adoption of machine translation, sentiment analysis, question-answering, and more (Vaswani et al., 2017; Devlin et al., 2018; Radford et al., 2018; Ouyang et al., 2022). Inspired by this, we can utilize the transformer backbone to preserve longer history information rather than the U-net backbone (Bao et al., 2022a; Shang & Ryoo, 2021). When utilizing a Transformer backbone, we observe that the model is comparatively more lightweight than a U-net backbone, with reduced training time and memory overhead. However, it is worth noting that since the Transformer requires positional encoding for sequences, and the training process of the diffusion model necessitates the inclusion of diffusion time step information, there may be interference between these two temporal aspects.

**Immediately-Conditional Sequence Generation.** Though we can use the diffusion model to plan a long-term sequence, only the first two states $s_t$ and $\hat{s}_{t+1}$ are adopted to produce actions with inverse dynamics $a_t = f_{inv}(s_t, \hat{s}_{t+1})$. Consequently, the direct influencing factor in obtaining rewards from the environment is the quality of the generated states $\hat{s}_{t+1}$. This enlightens us that we should pay more attention to the current generative state $\hat{s}_{t+1}$.

Immediate conditions are further categorized into two distinct types: those based on post-hoc filtering and those based on prior guidance. For the post-hoc filtering method, we can use state-reward sequences and filtrate high-quality sequences on the basis of multi-candidate sequences, while most previous works choose state sequences or state-action sequences for training. Prior guidance methods require reward sequence $\{r_t\}_{t:t+T-1}$ as guidance so as to instruct the generation process, i.e., reward sequence statistic from replay buffer is the most straightforward idea. Alternatively, reward regression (linear regression and quantile regression) is another choice. Although reward regression methods can offer a degree of extrapolation capabilities, they may also be prone to overfitting, resulting in significant estimation biases for OOD actions (Geman et al., 1992; Zhang et al., 2021). On the other hand, statistical reward methods directly utilize historical experience for guidance, which may, to some extent, constrain the model's performance (Bengio et al., 2009; Cobbe et al., 2019).

# F   MORE DISCUSSION ABOUT THE STRUCTURAL CHOICES

## F.1   INVERSE DYNAMICS

Following previous studies (Ajay et al., 2022; Janner et al., 2022), inverse dynamics is introduced to produce actions based on the state sequence generated by the diffusion model. We choose to model the distribution of state sequence rather than state-action sequence on the basis of two reasons: 1) The state is usually continuous in RL, but the mode of action is diverse, such as discrete and continuous in different environments. Separately modeling the state sequence makes the diffusion-based model more generic to extensive RL scenarios. 2) Usually, the action representation is high-frequency and less smooth, such as the joint torques in robotics control, which makes it more challenging to model and predict the action sequence.

## F.2   THE MAX DIFFUSION STEP

When we use the diffusion model for generation, we should choose the schedule of $\alpha_{1:K}$ and $\beta_{1:K}$ in advance. In our method, we choose the cosine beta schedule based on the previous studies (Nichol & Dhariwal, 2021; Wang et al., 2022b). However, it is important to note that this is not the sole option.

Table 11: The comparison of the original and accelerated sampling methods about the physical time consumption and the corresponding performance in Maze2D-umaze-v1 scenario for planning. The "O" refers to the "original version," and the "A" refers to the "accelerated version." Compared with the original version for generation, the accelerated sampling method greatly improves the sampling speed (**14.60×**) without significantly compromising performance.

| Diffusion time steps | 200(O) | 100(A) | 50(A) | 25(A) | 10(A) |
|---|---|---|---|---|---|
| Average planning time | 3.21 | 1.68 | 0.88 | 0.50 | 0.22 |
| Speed-up ratio | 1× | 1.91× | 3.65× | 6.42× | 14.60× |
| Mean performance | 126.8 | 120.1 | 119.2 | 119.4 | 118.8 |

Theoretically, we can opt for any diffusion time step. Yet, there is an inherent trade-off between the choice of diffusion time step and the performance of the model; a larger time step generally results in enhanced model generation (Ho et al., 2020; Rombach et al., 2022). Recently, many studies have been dedicated to investigating achieving as good performance as possible while minimizing the diffusion time step (Song et al., 2020; Bao et al., 2022b).

## G  DISCUSSION ABOUT LIMITATIONS AND FUTURE WORK

In terms of limitations, the mechanism of the generation process makes it slower than other models, such as Transformer-based models and MLP-based models, even though we can use recent break-throughs (Nichol & Dhariwal, 2021) to accelerate this process. More recent studies have brought hope for efficient generation (Song et al., 2023). Thus, we may be able to improve the generation efficiency based on advanced sampling methods, such as DDIM (Song et al., 2020). To show the potential of these accelerated technologies, we conduct the experiment on the Maze2D-umaze-v1 environment for planning with DDIM. In Table 11, we report the time consumption and performance of the original sequence generation version and the accelerated version. The results show that we can improve the sampling speed (**14.60×**) without significantly compromising performance. Another limitation is the restricted application on static datasets such as offline RL tasks because, in these static datasets, the joint distribution of samples is fixed. Applying diffusion-based RL methods to online RL tasks faces several challenges. The first challenge is the interaction cost that comes from the multiple generation steps. Fortunately, many studies try to improve the efficiency of diffusion methods during generation (Song et al., 2020; Nichol & Dhariwal, 2021). The second challenge is the slowly evolving data distribution, which implies that the diffusion model is no longer trained on a static dataset. The third is the issue of output stability in generative models, where the generated sequences might be overly sensitive to the current state, resulting in significant sequence output changes due to minor state variations.

