# OpenReview forum: "Instructed Diffuser with Temporal Condition Guidance for Offline Reinforcement Learning"
_ICLR.cc/2024/Conference — ICLR 2024 Conference Withdrawn Submission_

### Official Review · Reviewer_8Q3i · 2023-10-31

**Soundness:** 2 fair
**Presentation:** 2 fair
**Contribution:** 2 fair
**Rating:** 3
**Confidence:** 3

**Summary:**

This study explores temporal information within an offline reinforcement learning (offline-RL) setting for diffusion-based world models. Specifically, the paper presents three distinctive temporal conditioning strategies, prospective, historical and immediate conditions, to guide the diffusion sampling process. Building upon the Decision Diffuser, the authors model the distribution of state trajectories with the diffusion probabilistic model. The policy inference is implemented with a separated dynamic inverse module. The results on the D4RL benchmark demonstrated improved performance compared to the baseline methods. Ablation studies verified the effectiveness of the proposed methods.

**Strengths:**

- The illustrative examples provided in Figure 2 are insightful.
- Experiments on Gym-MuJoCo showed improvements, and the designed ablation studies verified the effectiveness of the proposed methods.

**Weaknesses:**

- What are the differences between the settings in Figure 2 (b) and (c)? It seems they both test the case where we have different reward right after the junction state and the same cumulative return after the junction state, i.e. they are both the immediate condition scenario.
- Could the authors also compare with Diffuser on the Maze2D tasks and explain the significant performance drop from Diffuser? What is the prospective condition used here? Also, could the authors clarify why the BC results in Table 2 differ greatly from those reported in the D4RL benchmark?
- To verify the effectiveness of the proposed method, could the authors please run some experiments on the sparse reward, high-dimensional state space tasks? i.e. the FrankaKitchen?
- Could the author also explain further the effects of historical sequence length L in Table 9, specifically what dose the noise refers to in the statement "In most other scenarios, the longer historical sequence (L = 10) provides negative improvements because the historical sequence may introduce useful information and extra noise concurrently, where the useless noise makes it hard for learning control."? Or would the performance drop due to the out-of-distribution generalization during test time? Since the longer the historical sequence is, the more likely the model has yet to see it during training.
- I think the immediate condition is not clear enough. What is the $\tau$ representation here? As in the previous description, $\tau$ is the state sequence. How is the reward $r_t$ modeled here?

**Questions:**

- Diffuser didn't use a dynamic inverse model, so the citation here is not accurate: "Following previous works (Janner et al., 2022; Ajay et al., 2022), we use inverse dynamics to produce actions based on the state sequence generated by the diffusion model because..."

---

### Official Review · Reviewer_kxUN · 2023-11-01

**Soundness:** 3 good
**Presentation:** 3 good
**Contribution:** 3 good
**Rating:** 5
**Confidence:** 3

**Summary:**

This paper considers the temporal conditions when applyting diffusion models on offline RL, and proposes Temporally-Composable Diffuser (TCD). TCD identifies three types of temporal conditions: historical, immediate, and prospective. All temporal conditons are incorporated as conditons of the diffusion model in a classifier-free manner. This paper also conducts numerous experiments, showing that TCD performs better against existing baselines and the designed conditions contribute to its good performance.

**Strengths:**

- The paper proposes a novel method for offline reinforcement learning using diffusion models and temporal conditions, which can capture the temporal dependencies of sequential data and achieve better controllable generation.
- The paper provides a comprehensive discussion and comparison of different temporal conditions and their effects on sequential generation, revealing potential improvements and insights for future work.
- The paper evaluates TCD on various offline RL tasks and shows that it outperforms or matches the state-of-the-art baselines, demonstrating the effectiveness and applicability of the proposed method.

**Weaknesses:**

See questions.

**Questions:**

1. Slight notation problems:
   - In the second paragraph of Sec. 3.1, please unify the notation of the mean of the generative process, $\mu_\theta(\tau^k,k)$ or $\mu_\theta(\tau^k)$.
   - Please use `\log`, which renders as $\log$, instead of $log$.
1. In Fig. 2, it seems that 2(b) and 2(c) are not well distinguished. Though there is explanation about immediat and prospective condtions in the main text, it is hard to read differences directly from these two subgraphs.
1. I did not find setting about the inverse dynamics model. Is it a pre-trained neural networks or some control theory-based models? I think the details of it should be included in the paper.
1. In Table 2, there is no standard deviation of DD or TCD, while all other tables report standard deviations. Moreover, I did not find your settings about how many independent runs are conducted to compute these means and standard deviations.
1. In Table 3, considering the standard deviations of the results, it is hard to say TCD is better  than TCD w/o $C_{IC}$ or TCD w/o $C_{HC}$, as there scores are too closed.

---

### Official Review · Reviewer_PzFh · 2023-11-02

**Soundness:** 3 good
**Presentation:** 3 good
**Contribution:** 3 good
**Rating:** 5
**Confidence:** 3

**Summary:**

In the context of offline RL, diffusion models are starting to be applied to model the sequences that make up RL tasks, like Decision Diffuser. The paper presents a diffuser-based method that take into account the past temporality of the sequence in the diffusion process to achieve better performance on the tasks.
The temporal aspect is added to the diffuser conditioning in 3 separate ways:
- prospective condition which plays the role of the return-based guidance. the proposed method uses return-to-go as opposed to q-value for better performance
- historical condition which is a novel inclusion
- immediate condition placing emphasis on the next action return
The method combining these aspects is called Temporally-Composable Diffuser (TCD), and is evaluated on several relevant benchmarks against relevant baselines including diffusion-model baselines.

**Strengths:**

- Including temporal information in the diffuser conditioning is an interesting idea that seem to be naturally suited to RL problems
- The baselines, datasets and comparisons chosen seem fair and extensive
- The difference between the main baseline the paper is built upon (DD) and TCD (presented method) is clearly highlighted
- The background section (3) clearly introduces the topic and benefits from diffusion models to RL, which makes the paper pretty self-contained, a remarkable feat given the space constraint
- I also found section 4 enlightening, where the authors lay out the benefits of the proposed formalism as questions readers can pose to themselves.

**Weaknesses:**

- The BC baseline is not defined
- The Related Works section explains the relationship between the presented method and other diffusion methods that are used as comparison baselines like DD, however some baselines could be better contextualised like MOReL, CQL and IQL.
- The RNN method mentioned in the RW section (Hafner et al) isn't used as a baseline without much explanation
- The reference section could benefit from showing the conference where a paper has been published instead of only the arxiv id, where applicable (e.g. the first reference).
- Although the paper is well-written, a few typos and awkward phrasing remain, like the use of "env" in the main text (end of section 4) or "regardless of which approach to be adopted", "equaling sequences"
- In section 5, $\bar{\epsilon}$, $\mathcal{C}$ and $\bar{\tau}$ could be better introduced
- I believe the Reward-To-Go acronym (RTG) isn't properly spelled out in the paper
- Improvements over DD are significant on 2 settings out of 9, which is nice but we are missing a tradeoff comparison: improvements even marginal are good if they provide low additional complexity, not so much if the added complexity is higher.

**Questions:**

- What is the relationship between MOReL, CQL, IQL and the presented method?
- Can you explain more clearly why the cited RNN method (Hafner et al 2019 & 2020) cannot be compared to your method on the chosen benchmarks?
- How exactly are the three presented conditionings C_PC, C_HC and C_IC combined? It seems to me this is an important piece of information for understanding the method that isn't clearly laid out in section 5 where it should be.
- I didn't fully understand the $\mathcal{C}_{HC}$ presentation in section 5. Are we replacing the closed-loop prediction history by ground truth history? How far are we looking?
- The presented method "only" significantly improves over DD on Walker2d-mr and Hopper-m tasks, while it is an interesting results, I worry that the presented method adds complexity to the diffuser conditioning for marginal improvements. Can you show that the method does not in fact add a lot of time and/or memory complexity over DD, which could highlight performance gains without much impact on training on inference times?

Overall the work is interesting but I need answers to these questions to be able to raise the final rating